# Design Requirements for Personal Mobility Vehicle (PMV) with Inward Tilt Mechanism to Minimize Steering Disturbances Caused by Uneven Road Surface

Tetsunori Haraguchi [1,2,*] and Tetsuya Kaneko [3]

1   Institutes of Innovation for Future Society, Nagoya University, Furocho, Chikusaku, Nagoya 464-8603, Japan
2   College of Industrial Technology, Nihon University, 1-2-1 Izumi-cho, Narashino 275-8575, Japan
3   Department of Mechanical Engineering for Transportation, Faculty of Engineering, Osaka Sangyo University, 3-1-1 Nakagaito, Daito 574-8530, Japan
*   Correspondence: haraguchi@nagoya-u.jp

**Abstract:** The Personal Mobility Vehicle (PMV), which has an inward-tilting angle, turns with lateral force due to a large camber angle, so it is necessary to consider the lateral movement of the tire vertical load axis during turning. Although the steering torque mechanisms are very different from those of automobiles, there are not many studies of the steering torque mechanisms of PMVs. In this paper, based on the effects of the force of six components acting on the tires, a method for setting the steering axis specifications is derived, including the geometrical minimization of steering moment disturbance due to the vertical load reaction force during turning. Automobile tires have a significant ground camber angle when traveling on rutted roads, but they do not have it on slanted roads because the vehicle body tilts along the road surface. On the other hand, in PMVs, the vehicle body always keeps upright when traveling both on slanted roads and on rutted roads. Therefore, the tires have ground camber angles on both types of road surface. We study the straight running ability under such road surface disturbances based on the geometrical minimization of steering moment disturbance due to the vertical load reaction force during turning. This straight running ability can be a remarkable strong point of PMVs with an inward tilt mechanism. In this study, it was proved that the steering axis parameters can be derived uniquely by taking into consideration the requirement to zero the moment (disturbance) around the steering axis due to the reaction force against the vertical load at all internal tilt angles.

**Keywords:** active tilting; steering axis; load reaction force; slanted road; rutted road





## 1. Introduction

Looking back to the beginning of the 20th century, in the early days of automobiles, there were already proposals for ultra-compact, low-cost mobility aimed at making automobiles affordable for ordinary people. Even after World War II, there were proposals for ultra-compact, low-cost mobility, mainly in the defeated countries of Japan, Germany, and Italy. It was a dream target for aircraft engineers because the development of aircraft was not permitted in the defeated countries. At the same time, it can be said that the aim was to make it affordable for the citizens of the defeated nations, due to their limited land and economic power, who could not afford large passenger cars, as could citizens from the United States.

Both of these first and second generations have in common that the top priority was to obtain the capacity for mobility at low cost, and at the same time compromising on the performance and satisfaction of car ownership. However, as the price of normal-sized cars fell and they became affordable for ordinary people, both generations of ultra-compact mobility were weeded out of the market, ending their product life.

However, the use of automobiles has improved people's lives with the progress of motorization after World War II, the rapid increase in the number of vehicles owned reduced their convenience due to traffic congestion and the lack of parking spaces in urban areas. In addition, since substantially all automobiles used fossil fuels as their energy source, in more recent years it has been pointed out that the $CO_2$ emitted by automobiles contributes significantly to global warming, following domestic consumption and industrial manufacturing.

In such a social situation, a new ultra-compact mobility concept called a personal mobility vehicle (PMV) is currently attracting attention. Among them, in recent years, aiming to save space compared to the first and second generations, a new concept with a fairly narrow width has been proposed [1–4]. The difference of third generation ultra-compact mobility compared with the first and second generations is not that it is inexpensive, but rather that it is "fun to drive" due to the inward-leaning vehicle dynamics and the electric motor. This will be a future-oriented new mobility concept.

In this way, the PMV is a promising concept for a sustainable mobility society, but compared to automobiles and motorcycles, there are still few examples of research on PMVs. The authors have been continuously conducting research on social acceptability and motion characteristics [3–14], mainly on PMVs with two front wheels and one rear wheel, which actively give an inward tilt angle according to the steering angle and the vehicle speed. In the References [5,6] the energy balance of the active inward-tilting mechanism was discussed as a basic study prior to the commercialization of the vehicle. Next, in the References [6–11], obstacle avoidance ability and inner wheel lift phenomenon during sudden steering were discussed as the points related to vehicle stability. In addition, in the References [12–14], as basic characteristics of vehicle dynamics, we have discussed the steady-state characteristics peculiar to PMVs that tilt inward when turning, the optimization of the gradient and hysteresis of steering torque characteristics, and the way to predict the turning limit.

When such ultra-compact and lightweight vehicles run in general traffic, they face more safety risks than automobiles, as do motorcycles. As described in the Reference [9] regarding obstacle avoidance ability, PMVs can be expected to be safer than automobiles in terms of active maneuverability on good road surfaces. However, from the crash safety point of view, we do not have a good answer yet. As shown in the reference [15], there are changes in reaction force against the vertical load due to unevenness on the road surface in the market, and there is concern that this disturbance may cause a dangerous situation such as lateral sway of the PMV. For this reason, there is a demand for a design method that prevents the PMV from swaying in the lateral direction or the steering pull against disturbances such as road surfaces and crosswinds when driving on the road.

## 2. Vehicle Characteristics, Design Parameters and Configuration of This Report

Research on stability against disturbances such as road surfaces and crosswinds has not progressed sufficiently yet, because the concept of PMV itself is fairly new.

Therefore, in this report, we consider the vehicle's susceptibility to external disturbances input from the road surface, such as left-right unequal changes in load reaction force (impact from the road surface) due to uneven road surfaces, and the inclination of the road surface in the transverse direction such as road slant and ruts. Then the design requirements for the steering axis of the PMVs, which the authors have been studying, are derived in order to maintain straight-line stability against such disturbances.

### 2.1. Vehicle Characteristics and Design Parameters

Focused Requirements on Steady Characteristic;

- Straight running ability on slanted roads (lateral vehicle movement and/or steering pull)
- Straight running ability on rutted roads (lateral vehicle movement and/or steering pull)

Design Parameters to Derive to Prevent the PMV from Lateral Vehicle Movement and/or Steering Pull;

- Tire camber characteristics (normalized camber stiffness ($D_y$), pneumatic trail by camber angle ($e_\gamma$))
- Front wheel steering axis (caster angle ($\xi$), caster trail ($T_\xi$), kingpin angle ($\psi$), kingpin offset ($D_\psi$))

*2.2. Configuration of This Report*

This report is a technical guideline for realizing such ultra-compact PMVs organized as follows.

1. Introduction
2. Vehicle Characteristics, Design Parameters and Configuration of this Report
   2.1. Vehicle Characteristics and Design Parameters
   2.2. Configuration of this Report
3. Materials and Methods
   3.1. Vehicle and Tire Specifications Used in this Report
       3.1.1. Vehicle Specifications
       3.1.2. Tire Specification
       3.1.3. Consideration of Generally Given Tire Camber Characteristic
   3.2. Steering Axis Design Parameters to Derive
       3.2.1. Steering Axis Design Parameters to Derive (Four Unknowns)
       3.2.2. Reduce the number of parameters to Derive (to Two Unknowns) by Previous Studies Related to Steering Axis Geometry
           1. Requirements for Caster Trail Considering Tire Lateral Force on the Road Surface [16]
           2. Kingpin Offset Requirements Considering the Longitudinal Braking Force in the Contact Surface on Straight Running
4. Results
   4.1. Derivation of Two Steering Axis Design Parameters from Two Requirements on Steady Characteristics
       4.1.1. A Method to Minimize Steering Disturbance Caused by Uneven Road Surface for Personal Mobility Vehicle (PMV) with Inward-tilting Mechanism
           1. Minimization of Steering Disturbance due to Reaction Force against the Vertical Load when Standing Upright
           2. Maintaining Zero Steering Disturbance due to Reaction Force against the Vertical Load even if the Tire Contact Point Moves Laterally when the Vehicle is Tilted Inward
       4.1.2. Specification Setting Procedure for Minimizing Steering Disturbance due to the Reaction Force against the Vertical Load
   4.2. Vehicle Stability during Disturbance Caused by Uneven Road Surface in the Market, Using the Method to Minimize Steering Disturbance
       4.2.1. Lateral Force Balance on Slanted Road
       4.2.2. Straight Running Stability on Slanted Road
       4.2.3. Straight Running Stability on Rutted Road
5. Discussion (Insight into Dynamic Phenomenon Analysis for the Future)
6. Conclusions

## 3. Materials and Methods

*3.1. Vehicle and Tire Specifications Used in This Report*

3.1.1. Vehicle Specifications

The object of this report is a PMV with an inward-tilting mechanism with two front wheels and one rear wheel. The static dimensional and mass specifications in Figure 1 and

Table 1 are the basic information for Sections 3.2 and 4. However, in Sections 3.2 and 4, we will try to optimize the design specifications as a general solution mainly from the static mechanical equilibrium conditions, so we do not need all the dimensions and mass specifications in the discussion. Unless otherwise noted in Sections 3.2 and 4, the inward-tilting mechanism does not matter whether it is passive or active. In the explanation of the previous study example in Section 3.2, note that an active inward-tilting mechanism is assumed in which the vehicle-tilting angle follows the target-tilting angle (Equation (1)) that is uniquely determined according to the vehicle speed and steering angle (target tracking control). The inertias in Table 1 will be used in future simulation experiments with multibody dynamics (MBD) models. This MBD model will also incorporate an active inward-tilting mechanism.

$$TRA = A \tan^{-1}\left(\frac{\sin(\delta)v^2}{lg}\right) \tag{1}$$

*TRA*: target roll angle
*A*: user amplification factor
*δ*: tire-steered angle
*v*: vehicle speed
*l*: wheel base

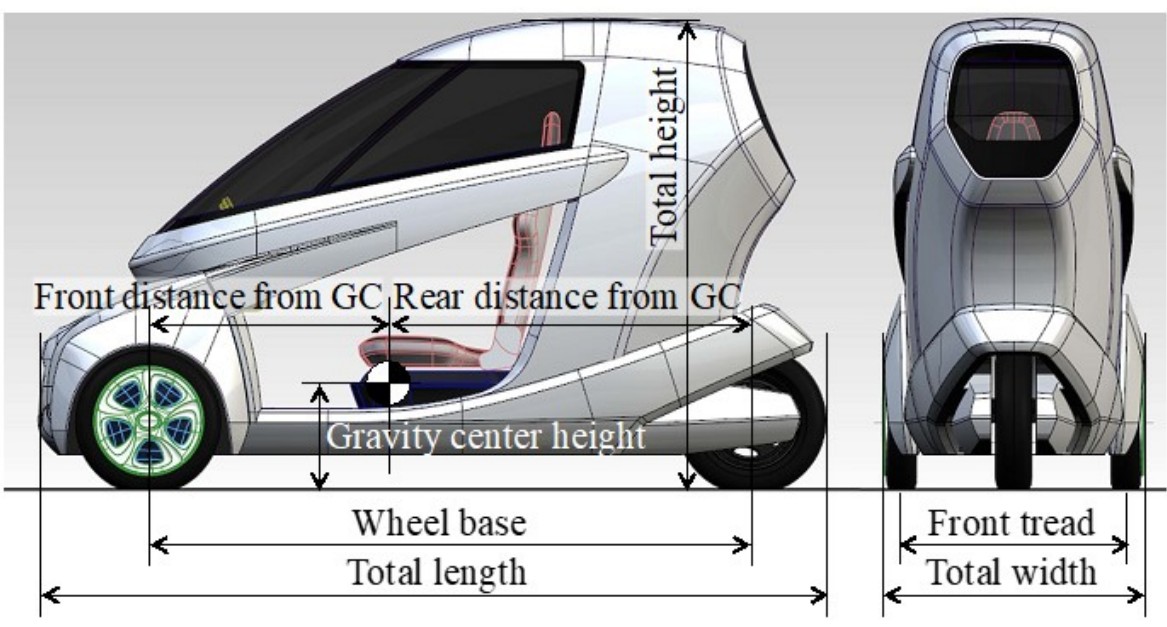

**Figure 1.** Dimensions of model vehicle [6].

**Table 1.** Specifications of model vehicle [6].

| Item | Unit | Value | Item | Unit | Value |
|---|---|---|---|---|---|
| Total length | m | 2.645 | Total mass | kg | 369.8 |
| Total width | m | 0.880 | Front mass distribution | kg | 222.1 |
| Total height | m | 1.445 | Rear mass distribution | kg | 147.7 |
| Wheel base | m | 2.020 | Roll moment of inertia | kgm$^2$ | 58.8 |
| Front distance from gravity center | m | 0.807 | (Roll moment of inertia of sprung mass) | kgm$^2$ | 43.0 |
| Rear distance from gravity center | m | 1.213 | Pitch moment of inertia | kgm$^2$ | 197.3 |
| Front tread | m | 0.850 | (Pitch moment of inertia of sprung mass) | kgm$^2$ | 118.0 |
| Gravity center height | m | 0.358 | Yaw moment of inertia | kgm$^2$ | 187.3 |
| Steering gear ratio | - | 16.0 | (Yaw moment of inertia of sprung mass) | kgm$^2$ | 102.3 |

### 3.1.2. Tire Specifications

The specifications of the 100/90ZR12 size motorcycle tires used in this report are shown in Figures 2 and 3, Table 2, which are same as the References [4–14,16]. The treads of motorcycle tires are characterized by a substantially circular cross-section compared to automobile tires, because they are used with a large camber angle. The cross-sectional shape in Figure 2 corresponds to the value when the tire is properly inflated and no load is applied, and these dimensional specifications become indispensable when considering the lateral movement of the tire contact point when the vehicle is tilted inward in Sections 4.1.1 and 4.2. For the sake of simplicity, this report does not consider the elastic deformation of the tire in each direction. Motorcycle tires do not deform much even when under load, therefore, when deriving the steering axis specifications from the viewpoint of geometric vehicle mechanisms, it is assumed that there is no error using tire dimensions under no load. The tire characteristic map in Figure 3 is used in Section 3.2.2 and will be used in future dynamic simulations. The tire characteristic coefficients in Table 2 are used in the discussion of static balance in Sections 3.2 and 4; however, we do not need all coefficients in the discussion of this report, as with the dimensional and mass specifications of the vehicle.

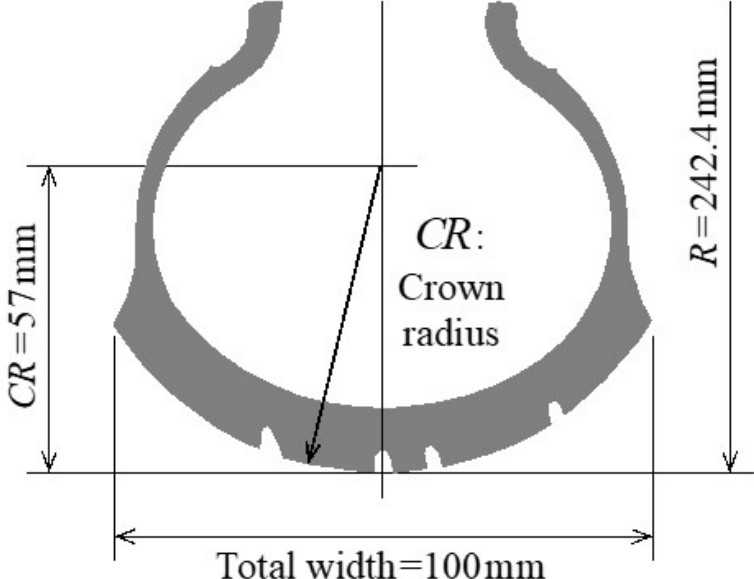

**Figure 2.** Typical tire cross section (Motorcycle, PMV) [16].

### 3.1.3. Consideration of Generally Given Tire Camber Characteristics

In a motorcycle, unlike an automobile, the centripetal force required for turning is obtained mainly by the camber angle of the tires to the ground (camber thrust), then the tires do not have a large slip angle.

The normalized camber stiffness ($D_y$) is obtained by dividing the camber thrust by the camber angle and the vertical load. Therefore, at $D_y$ (/rad) $\approx$ 1 or $D_y$ (/deg) $\approx \pi/180 = 0.0174$, the camber thrust alone provides the centripetal force necessary to turn. Table 2 shows that such characteristics are generally given to motorcycle tires.

Considering the lateral movement of the contact point due to the camber angle to the ground, the normalized camber stiffness to obtain the centripetal force required for turning by the camber thrust alone is $D_y$ (/rad) < 1. This will be discussed later also in Section 4.2.

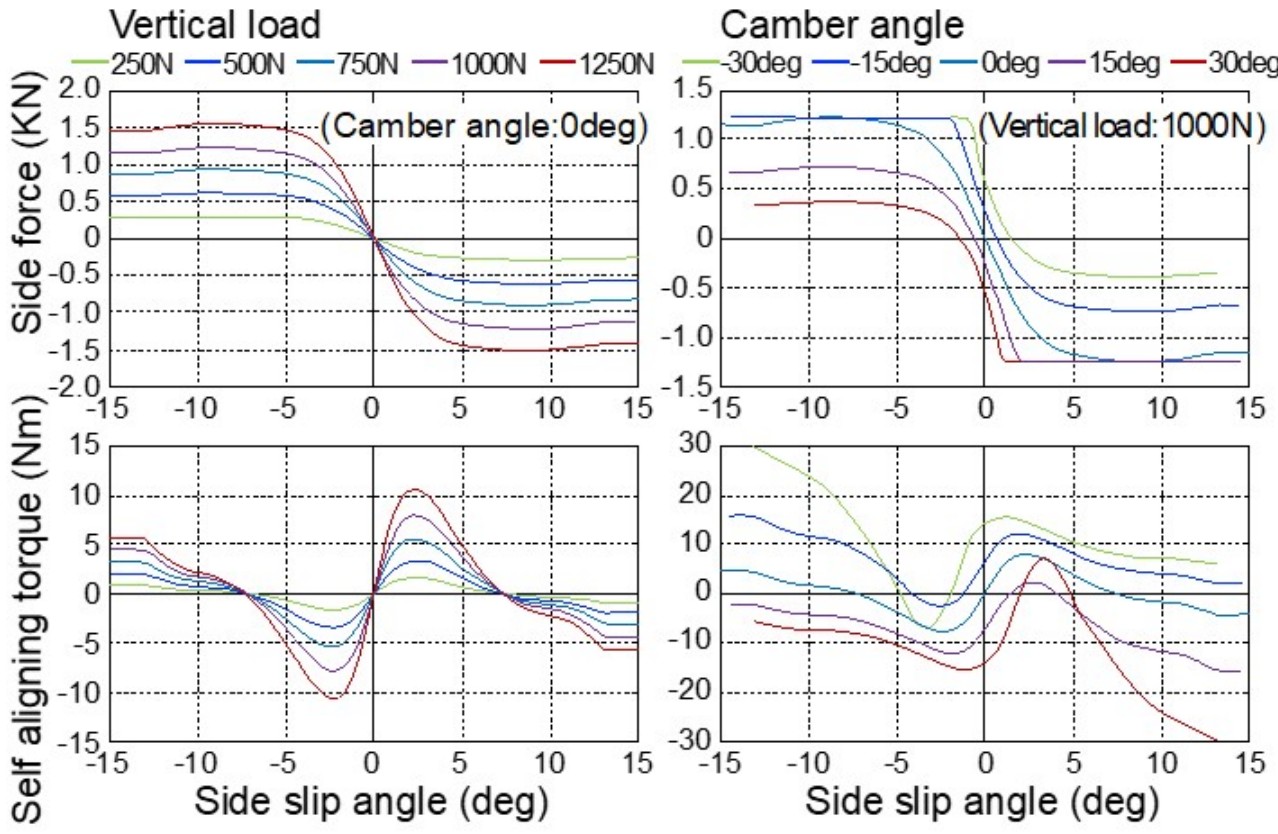

**Figure 3.** Motorcycle tire model is used in CarMaker [13].

**Table 2.** Parameters of Tire (100/90R12), Vertical load: $F_z$ = 1000N.

| Item | Value | Item | Value |
|---|---|---|---|
| Tire radius: $R$ | 0.242 m | Crown radius: $CR$ | 0.057 m |
| Cornering stiffness: $K_y$ | 427.4 N/deg | Camber stiffness: $Q_y$ | 17.77 N/deg |
| Normalized $K_y$: $C_y$ | $42.74 \times 10^{-2}$/deg | Normalized $Q_y$: $D_y$ | $1.777 \times 10^{-2}$/deg |
| Pneumatic trail on slip angle: $e_\beta$ | 0.0136 m | Pneumatic trail on camber angle: $e_\gamma$ | −0.0267 m |

### 3.2. Steering Axis Design Parameters to Derive

3.2.1. Steering Axis Design Parameters to Derive (Four Unknowns)

To uniquely define an axis in three-dimensional space, it is necessary to define six parameters for two points, or one point and one direction vector. In terms of determining the steering axis uniquely in the suspension design, this means determining four specification values other than the two specification values of the longitudinal position and tread of the front wheels.

The four independent parameters are caster angle ($\xi$), kingpin angle ($\psi$), caster trail ($T_\xi$) and kingpin offset ($D_\psi$) at ground level as shown in Figure 4. Alternatively, the tire radius ($R$) can be used to set $T_{off} = T_\xi - R\sin\xi$ and $D_{off} = D_\psi - R\sin\psi$ at the height of the wheel axis as independent values.

As a previous study on automobile suspension at the moment around the steering axis generated by the tire force on the road surface and the reaction force of the load, there are examples. The authors discussed the steering pull phenomenon during braking on a rutted road, focusing on the lateral movement of the load center on the ground [17,18], and the authors also discussed the vehicle drift on slanted roads considering tire characteristics near neutral [19]. However, there is no example that discusses the steering torque characteristics of PMV.

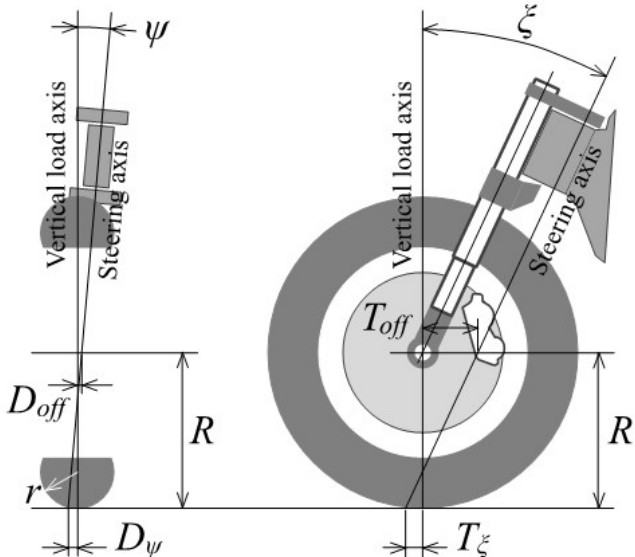

$\xi$: Caster angle
$T_\xi$: Caster trail
$T_{off}$: Caster trail at wheel center height
$\psi$: Kingpin angle
$D_\psi$: Kingpin offset
$D_{off}$: Kingpin offset at wheel center height

**Figure 4.** Steering axis and four independent parameters.

In addition, as an example of a previous study on PMVs that tilt inward when turning, the authors proposed a method to optimize steering torque characteristics on the pulled side [12]. There are also examples of motorcycles that discuss the directional stability using the camber angle [20] and that discuss the steering system torsional stiffness [21].

In order to uniquely derive the four specification values of the steering axis, it is necessary to determine the four target characteristic requirements for setting the steering axis. Based on the requirements of steering torque when a lateral force is applied to the tire on the road surface [12] and the requirement of straight-line stability when a longitudinal force is applied, we newly add the requirements for minimizing the disturbance around the steering axis due to the reaction force of the vertical load from road surface. In order to satisfy the target number of requirements, the requirement when the vehicle is upright and the requirement when the vehicle is tilted are taken up as two independent target requirements [16].

3.2.2. Reduce the Number of Parameters to Derive (to Two Unknowns) by Previous Studies Related to Steering Axis Geometry

1. Requirements for Caster Trail Considering Tire Lateral Force on the Road Surface [16]

According to the Reference [12], in case of PMVs that tilt inward when turning, the lateral force for turning is mostly obtained by the camber thrust and the moment when the pulled side is generated. In general, such torque characteristics are steered by the bar-handle of motorcycles, and there are no examples of steerability by the steering wheel such as in a car. For this reason, it seems difficult for the driver of the PMV who steers with the steering wheel to control the steering moment in the pulled side. Therefore, canceling this moment is the requirement for the caster trail ($T_\xi$).

In addition, with PMVs that actively provide an inward tilt angle according to the steering angle and vehicle speed, an inadvertent large steering angle input on a low $\mu$ road induces an excessive inward inclination, creating the risk of overturning. In addition, counter steering during oversteering causes a dangerous outward tilt and induces trip over. In order to prevent these risks, it is necessary to avoid also the steering torque on the returning side due to excessive $T_\xi$.

In the References [12,13], as shown in Equation (2) and Figure 5a,b, it is proposed that the caster trail ($T_\xi$) offsets the pneumatic trail ($e_\gamma$) caused by the camber angle; the additional torque proportional to the roll angle of the vehicle is added to the steering wheel in order to offset the hysteresis of the steering torque; and finally the additional torque proportional to the prospected lateral acceleration of the vehicle, proportional to the square

of the vehicle speed and the steering wheel angle, is added to obtain an appropriate slope of the steering torque. The appropriate value of the added torque, which is proportional to the prospected lateral acceleration, should be determined based on the human sense by driving the vehicle. However, this value is not related to this report. It is only concerned to offset the pneumatic trail ($e_\gamma$) due to the camber angle with the caster trail ($T_\xi$).

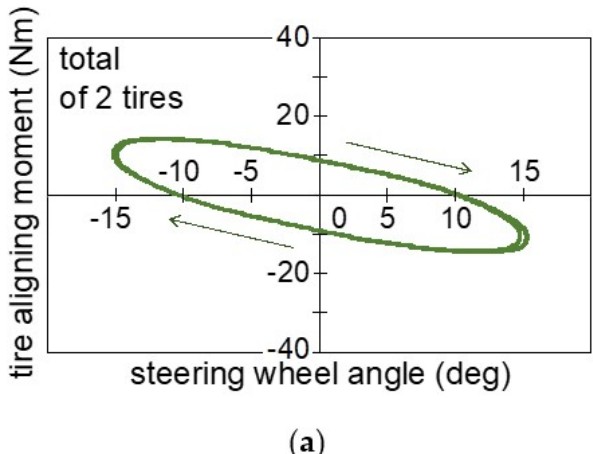 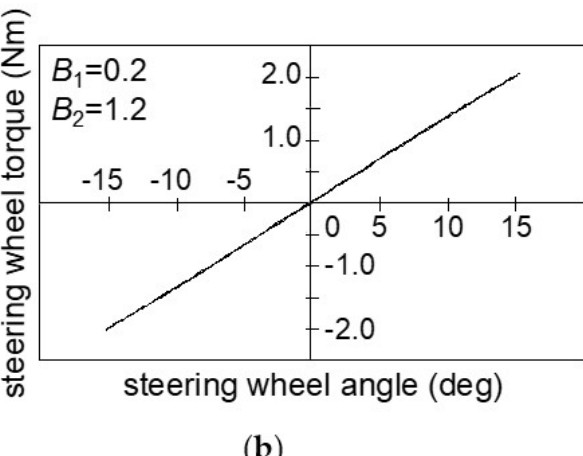

(a)                                                           (b)

**Figure 5.** (**a**) Original tire aligning moment. (**b**) Adjusted steering wheel torque.

Therefore, based on the idea proposed in the Reference [12], we set $T_\xi$ to be equal to the absolute value of the pneumatic trail ($e_\gamma$) due to the camber angle ($\gamma$) ($T_\xi$ = 26.7mm) also in this report and proceed with the following study.

$$MT = \frac{(M_z + F_y(-e_\gamma))}{r_s} + B_1 PLA + B_2 \varphi \tag{2}$$

$MT$: steering wheel torque
$M_z$: tire aligning moment
$B_1$: aligning torque adjustment factor
$F_y$: lateral force
$PLA$: provisional lateral acceleration
$e_\gamma$: tire pneumatic trail on camber angle
$B_2$: hysteresis adjustment factor
$r_s$: steering ratio
$\varphi$: tilt angle

2. Kingpin Offset Requirements Considering the Longitudinal Braking Force in the Contact Surface on Straight Running

As mentioned in the previous subsubsection, there is a proposal in the Reference [12] regarding lateral force input. However, there is no example of a study that assumes PMV of the front two wheels for longitudinal force. Ideally, the vehicle should go straight during one-side braking even without the hand on steering wheel. For this purpose, the steering axis is placed outside the center of the ground point of the tire. As shown in Figure 6b, the difference of toe-in moment due to the difference of braking force causes the front wheels to steer to the smaller braking force side, and the yaw moment due to the lateral forces offsets the yaw moment disturbance due to the unbalance of braking forces [16].

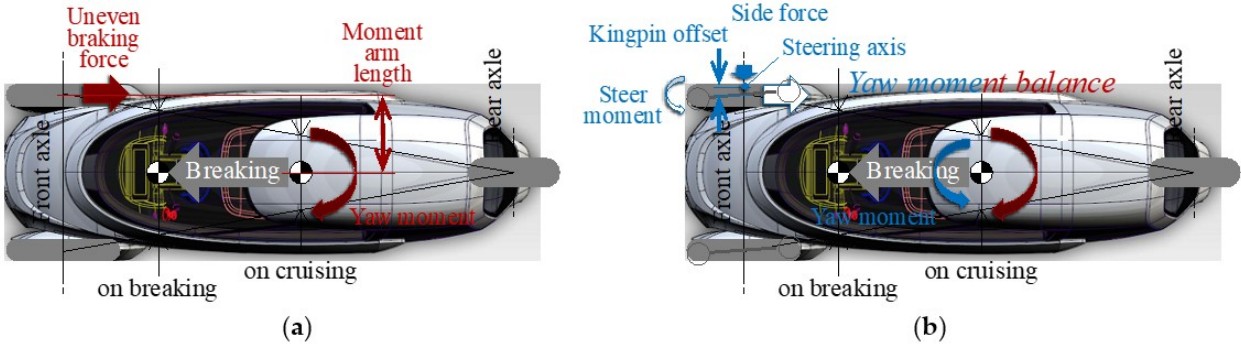

**Figure 6.** (**a**) Yaw moment on uneven braking force. (**b**) Yaw moment balance on uneven braking force.

This negative kingpin offset ($D_\psi$) is the requirement to keep straight running, as shown in the Equation (3), obtained by the simplified moment around the steering axis assumed to be nearly vertical due to the longitudinal force on the ground surface. This condition is shown in Figure 7. In automobiles, it is a constant value of about −21mm, and in motorcycles, it is always 0 mm. However, in the case of PMVs with active inward-tilting mechanism, an inward-tilting angle that is proportional to the square of the vehicle speed and the steering angle is inevitably generated from Equation (1). Since a proportional lateral force by tire camber angle is generated, the requirement to run straight on one-side braking is the kingpin offset value, which is dependent on vehicle speed, as shown in Figure 7.

$$D_\psi \approx \frac{C_y Tr \left( T_\xi + e_\beta \right)}{2l_f \left( C_y + D_y \frac{v^2}{lg} \right)} \tag{3}$$

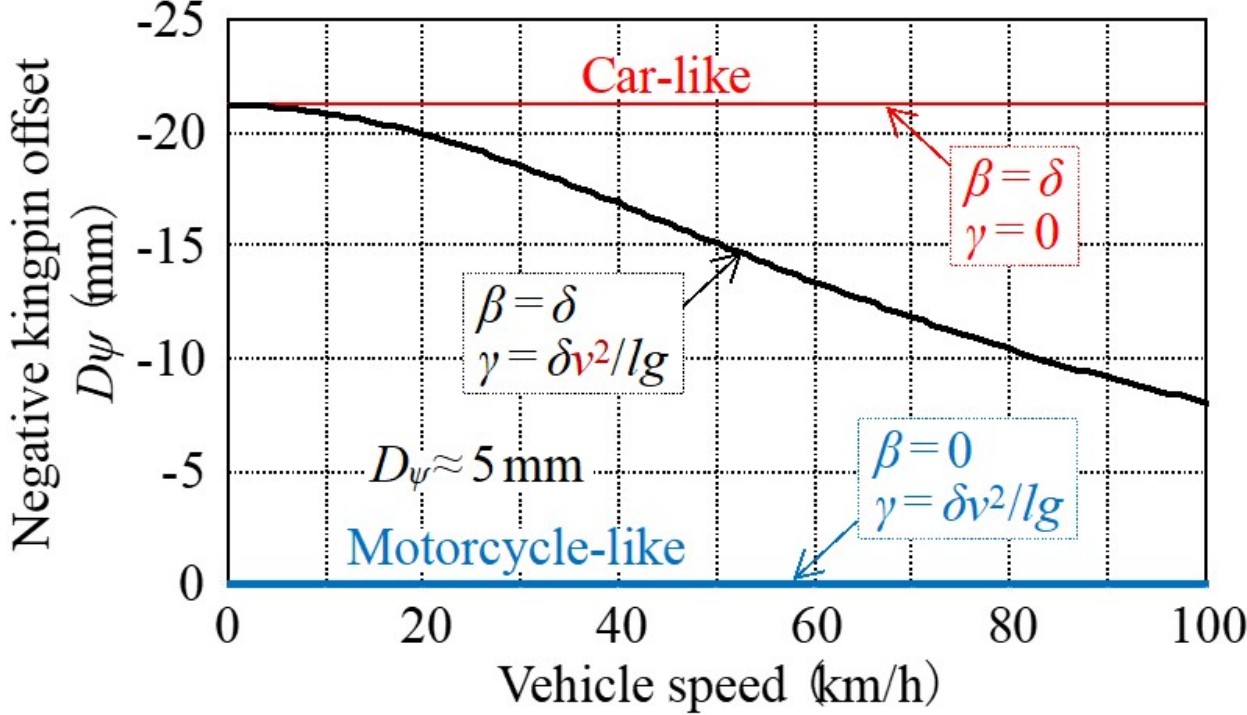

**Figure 7.** $D_\psi$ to minimize yaw moment disturbance by one-side braking.

However, looking at the example of the kingpin offset value of a passenger car with a sufficient market experience, it is actually not a large negative value such as −21mm, but a modest value of −5mm at most [17,18]. This indicates that the requirements for straight

running in the market for hands-free driving do not match the situation. As long as the driver holds the steering wheel, a large kingpin offset is not necessary, and an excessive kingpin offset rather induces unpleasant rotational vibration of the steering wheel when ABS is activated. It is thought that this modest kingpin offset value was determined as a result. Since the market suitability of PMV is considered to be the same, regardless of the Equation (3) and Figure 7, this report sets the kingpin offset value at −5mm, following the example of passenger cars, and will proceed with further studies.

## 4. Results

*4.1. Derivation of Two Steering Axis Design Parameters from Two Requirements on Steady Characteristics*

4.1.1. A Method to Minimize Steering Disturbance Caused by Uneven Road Surface for Personal Mobility Vehicles (PMVs) with Inward-Tilting Mechanism

Of the four independent parameters described in Section 3.2; caster angle ($\xi$), kingpin angle ($\psi$), caster trail ($T_\xi$), and kingpin offset at ground level ($D_\psi$), caster trail ($T_\xi$) and kingpin offset ($D_\psi$) have already been assumed according to Section 3.2, therefore, in this section the caster angle ($\xi$) and kingpin angle ($\psi$) are derived from the two target characteristics of minimizing steering disturbance due to the reaction force against the vertical load when standing upright, and of maintaining zero steering disturbance due to the reaction force against the vertical load even if the tire contact point moves laterally when the vehicle is tilted inward.

1.  Minimization of Steering Disturbance due to Reaction Force against the Vertical Load when Standing Upright

As shown in Figure 8, if the point on the ground on the steering axis is $P_1(x_1, y_1, 0)$, the other point is $P_2(x_2, y_2, z_2)$, and $|P_1 - P_2| = 1$, the four parameters $\xi$, $\psi$, $T_\xi$, and $D_\psi$ are represented by four coordinate values $x_1$, $y_1$, $x_2$, and $y_2$ as follows.

$$tan\xi = (x_2 - x_1)/z_2 \quad \xi : Caster\ angle$$
$$tan\psi = (y_2 - y_1)z_2 \quad \psi : Kingpin\ angle$$
$$T_\xi = -x_1 \quad T_\xi : Caster\ trail$$
$$D_\psi = -y_1 \quad D_\psi : Kingpin\ offset$$
$$z_2^2 = 1 - (x_2 - x_1)^2 - (y_2 - y_1)^2$$

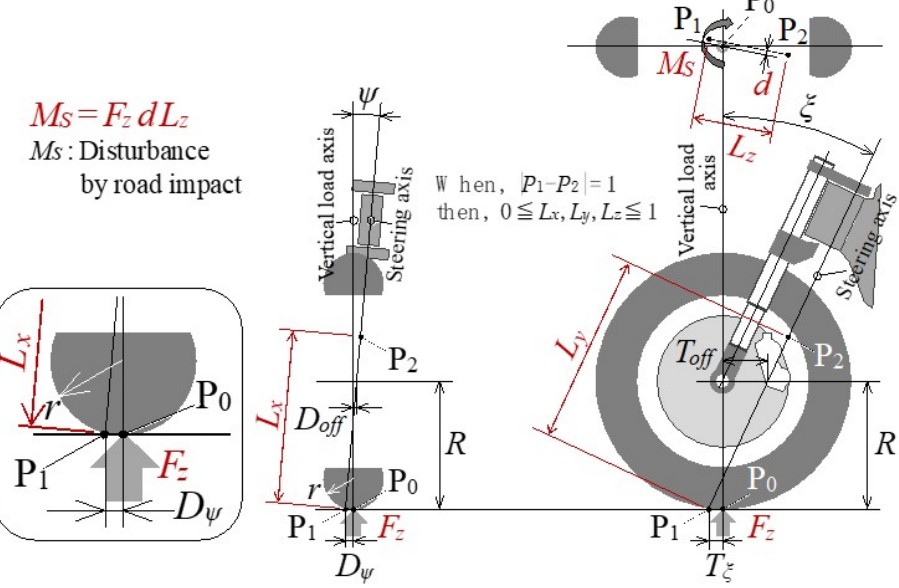

**Figure 8.** Steering moment disturbance ($M_S$) by road impact ($F_z$).

Here, the projection length $L_z$ to the *x-y* plane, the projection length $L_x$ to the *y-z* plane, and the projection length $L_y$ to the *z-x* plane of the line segment $P_1 - P_2$ are as follows.

$$
\begin{aligned}
L_x^2 &= (y_2 - y_1)^2 + z_2^2 = 1 - (x_2 - x_1)^2 \\
L_y^2 &= (x_2 - x_1)^2 + z_2^2 = 1 - (y_2 - y_1)^2 \\
L_z^2 &= (x_2 - x_1)^2 + (y_2 - y_1)^2
\end{aligned}
\tag{4}
$$

From $|P_1 - P_2| = 1$, $L_x$ is the efficiency of the moment around the steering axis created by the vertical force and by the longitudinal force, $L_y$ is by the lateral force, and $L_z$ is by the efficiency. $L_x$, $L_y$ and $L_z$ are also represented by four coordinate values $x_1$, $y_1$, $x_2$, and $y_2$.

The reaction force axis is defined by the vertical axis passing through the center of the ground point $P_0(x_0, y_0, 0)$ in Figure 8. This reaction force axis and the steering axis are generally in a skewed position. Since the reaction force axis is vertical, the distance between the two axes is equal to the distance $d$ between the point $P_0$ and the steering axis $P_1 - P_2$ in planar view as shown in Figure 8.

If $P_1 - P_2$ in planar view is set to $ax + by + c = 0$,

$$
\begin{aligned}
a &= y_2 - y_1 \\
b &= -(x_2 - x_1) \\
c &= x_2 y_1 - x_1 y_2
\end{aligned}
$$

$d$ is expressed as;

$$
d^2 = (ax_0 + by_0 + c)^2 / (a^2 + b^2)
\tag{5}
$$

Also, Equation (4) can be transformed into;

$$
L_z^2 = a^2 + b^2
\tag{6}
$$

Here, the moment $M_S$ around the steering axis due to the reaction force against the vertical load $F_z$ is expressed by Equation (7) as the product of these.

$$
M_S = F_z d L_z
\tag{7}
$$

When the vehicle is upright, the coordinates of the center of the ground point $P_0$ are $(0, 0, 0)$ and $d$ is $d^2 = c^2 / (a^2 + b^2)$. By substituting Equations (5) and (6) into Equation (7), the moment around the steering axis ($M_S$) becomes Equation (8).

$$
M_S = F_z d L_z = F_z C = F_z (x_2 y_1 - x_1 y_2)^{1/2}
\tag{8}
$$

The condition of $M_S = 0$ is shown in Equation (9) by modifying $c = x_2 y_1 - x_1 y^2 = 0$, and Figure 9 is obtained by substituting $T_\xi = 26.7$ mm and $D_\psi = 5$ mm.

$$
\frac{y_1}{x_1} = \frac{y_2}{x_2} = \frac{D_\psi}{T_\xi} = \frac{tan\psi}{tan\xi}
\tag{9}
$$

Expressing $d = 0$ graphically, as shown in Figure 10, it is a special state where the reaction force axis and the steering axis intersect at the point $P_h$. If the coordinates of $P_h$ are $(0, 0, h)$, $h$ is expressed by Equation (10).

$$
h = \frac{T_\xi}{tan\xi} = \frac{D_\psi}{tan\psi}
\tag{10}
$$

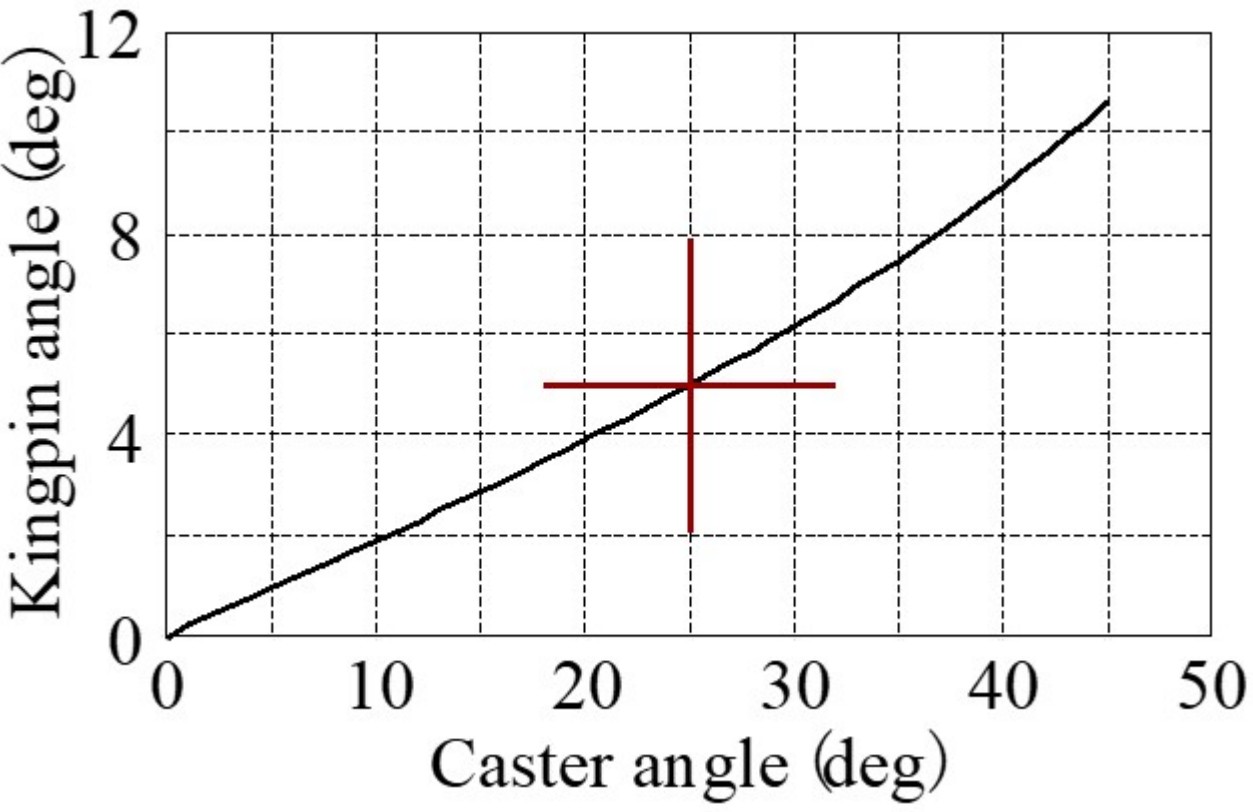

**Figure 9.** Steering axis to minimize $M_s$ by $F_z$ on zero roll condition.

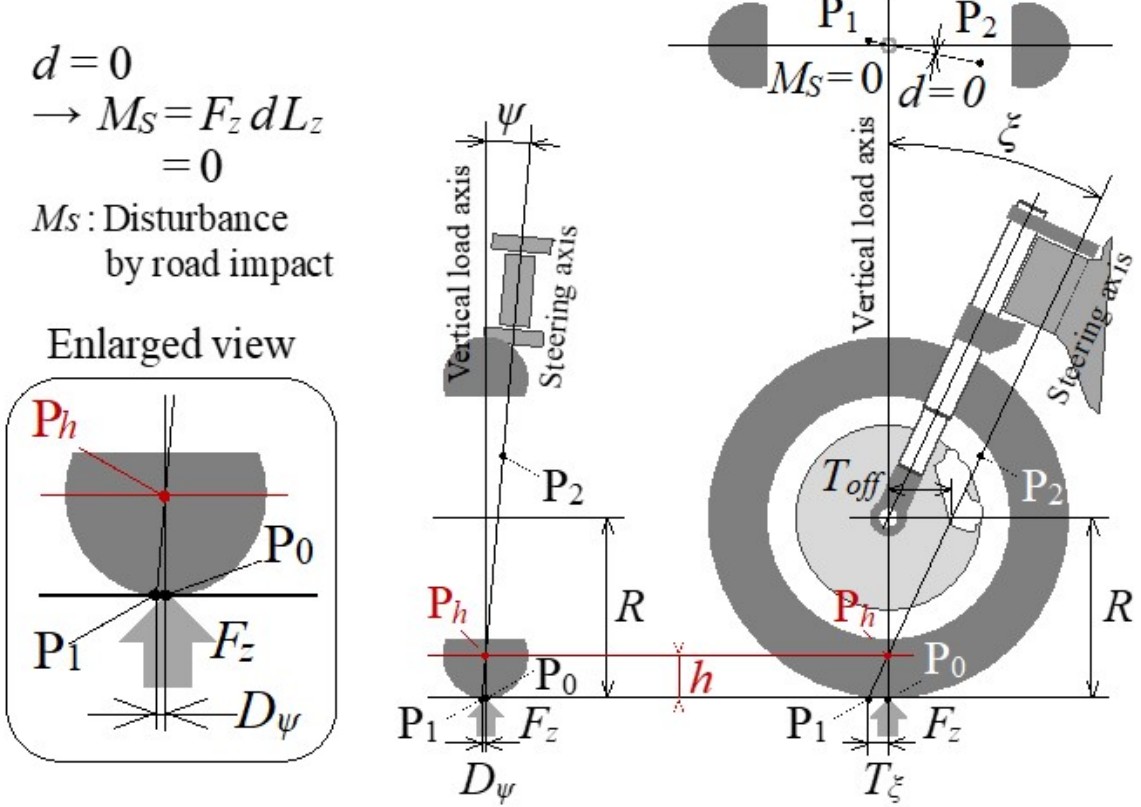

**Figure 10.** Steering axis to minimize $M_s$ by $F_z$ on upright condition.

2.    Maintaining Zero Steering Disturbance Due to Reaction Force against the Vertical
      Load Even If the Tire Contact Point Moves Laterally When the Vehicle Is Tilted Inward

Figure 11 shows the lateral movement of the tire contact point, and the reaction force when the tire is tilted inward, with the crown radius ($r$) enlarged for clarity. When the vehicle body tilts by an angle $\varphi$, the steering axis rotates with the tires. Points $P_0$, $P_1$, $P_2$ move to $P_0^*$, $P_1^*$, $P_2^*$.

$$P_0(0,\ 0,\ 0),\ P_1(x_1,\ y_1,\ 0),\ P_2(x_2,\ y_2,\ z_2)$$
$$x_1 = -T_\xi,\ \ x_2 = x_1 + L_y sin\xi$$
$$y_1 = -D_\psi,\ \ y_2 = y_1 + L_x sin\psi$$
$$z_2 = L_x cos\psi = L_y cos\xi$$
$$P_0^*(0,\ y_0^*,\ z_0^*) = (0,\ r(\varphi - sin\varphi),\ r(1 - cos\varphi))$$
$$P_1^*\left(x_1,\ y_1^*,\ z_1^*\right) = \left(x_1,\ y_0^* - D_\psi cos\varphi,\ z_0^* + D_\psi sin\varphi\right)$$
$$P_2^*(x_2,\ y_2^*,\ z_2^*) = \left(x_2,\ y_1^* - L_x sin(\varphi + \psi),\ z_1^* + L_x cos(\varphi + \psi)\right)$$

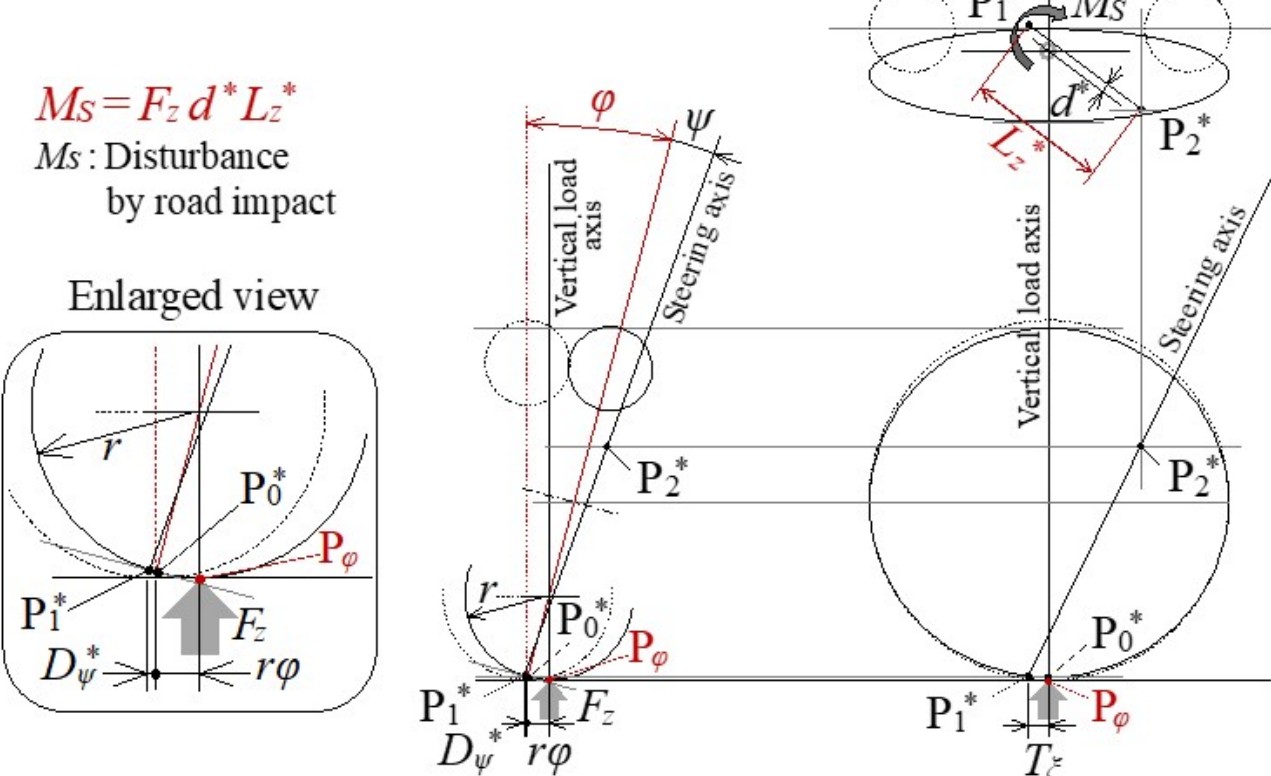

**Figure 11.** Steering axis on tilted condition. The * shows when the tire is tilted inward.

As shown in Figure 11, the tire contact point $P_\varphi$ moves from $P_0$ to the inner side of the vehicle by $r\varphi$, and becomes $P_\varphi(0, y_\varphi, 0) = (0, r\varphi, 0)$. Assuming that the steering axis $P_1^* - P_2^*$ during tilt is $a^*x + b^*y + c^* = 0$,

$$a^* = y_2^* - y_1^*$$
$$b^* = b = -(x_2 - x_1)$$
$$c^* = x_2 y_1^* - x_1 y_2^*$$

$d^*$ is expressed as

$$d^{*2} = (br\varphi + c^*)^2 / \left(a^{*2} + b^2\right)$$

Also, the projection length $L_z^*$ of the line segment $P_1^* - P_2^*$ onto the *x-y* plane is as follows.

$$L_z^{*2} = (x_2 - x_1)^2 + (y_2^* - y_1^*)^2 = a^{*2} + b^2$$

Here, the condition for $M_s = F_z d^* L_z^* = F_z(br\varphi + c^*) = 0$ is $br\varphi + c^* = 0$.

For the sake of simplicity, if we graphically consider the state where the reaction force axis and the steering axis intersect, the coordinates of the intersection point $P_h^*$ are (0, $r\varphi$, *h*), where *h* corresponds to *r* shown as Figure 12. Equation (11) is obtained regardless of $\varphi$.

$$r = h = \frac{T_\xi}{tan\xi} = \frac{D_\psi}{tan\psi} \tag{11}$$

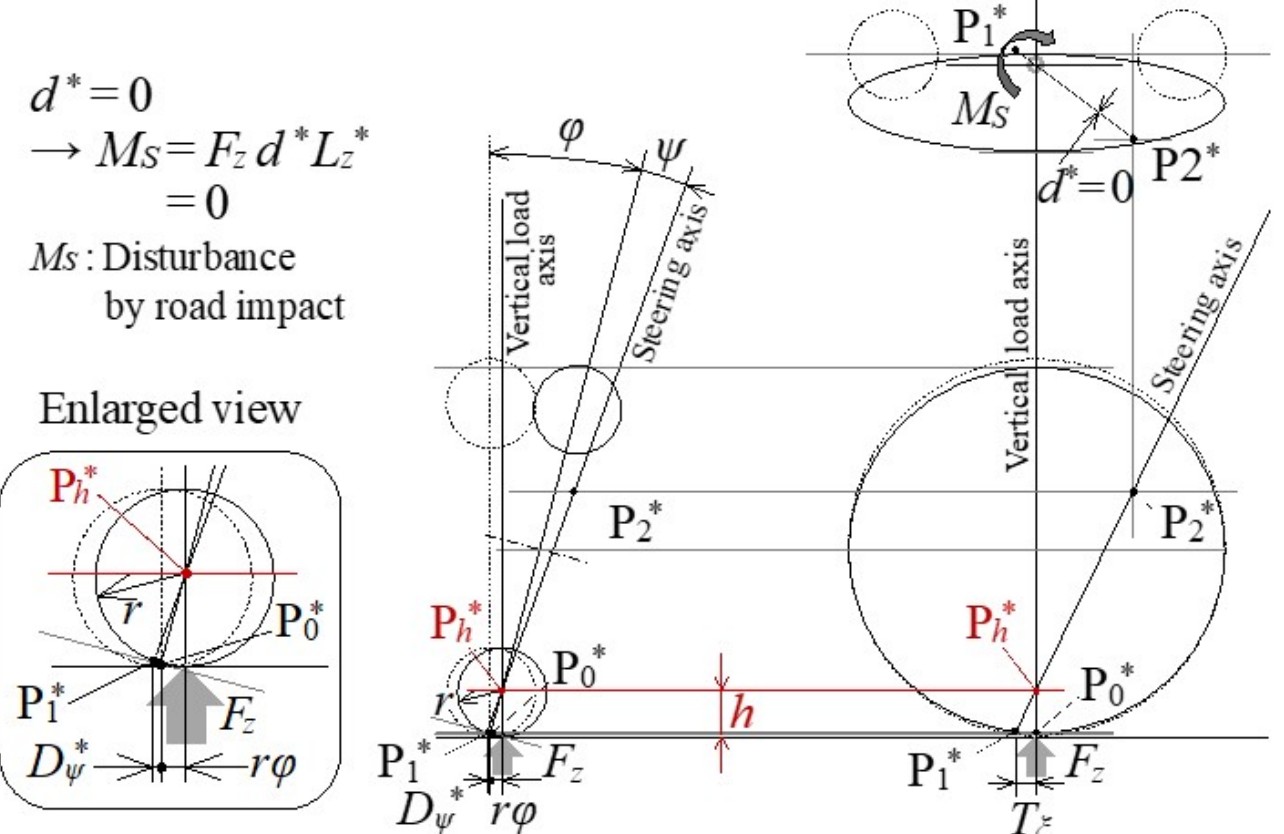

**Figure 12.** Steering axis to minimize $M_s$ by $F_z$ on tilted condition. The * shows when the tire is tilted inward.

4.1.2. Specification Setting Procedure for Minimizing Steering Disturbance Due to the Reaction Force against the Vertical Load

When planning a vehicle, the tire size is determined according to its load capacity based on vehicle mass and maximum speed requirements. If the size is too small, there is concern that the load capacity will be insufficient. If the size is too large, the unsprung road trackability deteriorates and the grip of the tire becomes unstable. In other words, the appropriate tire size is almost uniquely determined according to the vehicle size, and the crown radius (*r*) is also almost uniquely determined.

Figure 13 shows the procedure for setting the steering axis specifications when the crown radius (*r*) is determined, as a summary of the method for minimizing the disturbance around the steering axis due to the reaction force against the vertical load. Assuming $T_\xi = 0.0267$ m from Section 3.2.2 and $D_\psi = 0.005$ m from Section 3.2.2.

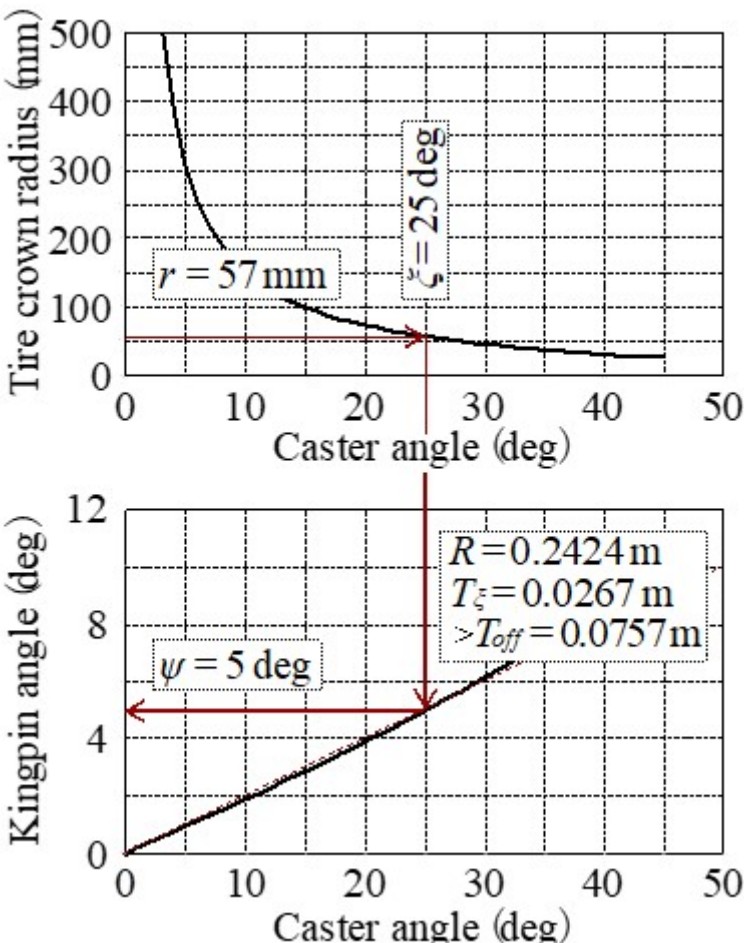

**Figure 13.** Procedure to set the steering specifications.

(1)  Caster angle ($\xi$) = 25 deg is obtained from $r$ = 0.057 m in order to maintain the $d$ value during tilt.
(2)  In order to set $d$ = 0 when standing upright, the kingpin angle ($\psi$) = 5 deg can be obtained from $\xi$ = 25 deg.

Using the tire radius ($R$) = 0.2424 m, the offset of the steering axis at the height of the wheel axis is as follows.

$$T_{off} = T_\xi - Rsin\xi = 0.0267 - 0.2424sin(25\ \text{deg}) = -0.0757\ \text{m}$$
$$D_{off} = D_\psi - Rsin\psi = 0.005 - 0.2424sin(5\ \text{deg}) = -0.0161\ \text{m}$$

*4.2. Vehicle Stability during Disturbance Caused by Uneven Road Surface in the Market, Using the Method to Minimize Steering Disturbance*

As an application of maintaining zero steering disturbance due to the reaction force against the vertical load even if the tire contact point moves laterally when the vehicle is tilted inward, first consider the state of maintaining straight running on a slanted road surface with a transverse slope. Next, the ability to maintain straight running on rutted roads, which are often encountered in the market along with slanted roads, is also considered.

4.2.1. Lateral Force Balance on Slanted Road

In a vehicle without an inward-tilting mechanism, the lateral force that prevents the vehicle from sliding down along the road surface is generated by the slip angle of all wheels caused by the vehicle slip angle, as shown in Figure 14. Assuming that the lateral transfer

of vertical load between both wheels and the slip angle due to slant are sufficiently small, and that there is no front wheel steered angle, the lateral force balance is expressed by Equation (12), and the required slip angle ($\beta$) is obtained by Equation (13). In order to generate this slip angle ($\beta$), a yaw angle of angle ($\beta$) is generated in the vehicle body. The yaw moment balance is expressed by Equation (14), but this equation is nothing more than the indication of the vehicle's center of gravity (G.C.) position (Equation (15)). In the transition section from a road surface without slant to a road surface with slant angle ($\varphi$), dynamic response is affected by yaw inertia ($I_z$) and roll inertia ($I_x$). However, it transitions relatively quickly to the balanced state above, under the condition both $\beta$ and $\varphi$ are small.

$$\beta_r = \beta_{fL} = \beta_{fR} = \beta \; F_{y\beta r} \approx K_{yr}\beta = C_y F_{zr}\beta$$
$$F_{y\beta fL} \approx K_{yfL}\beta = C_y F_{zfL}\beta \; F_{y\beta fR} \approx K_{yfR}\beta = C_y F_{zfR}\beta$$
$$mg\sin\varphi = F_y = F_{y\beta r} + F_{y\beta fL} + F_{y\beta fR} = C_y\beta\left(F_{zr} + F_{zfL} + F_{zfR}\right) = mgC_y\beta \tag{12}$$

$$\beta = \sin\varphi/C_y \approx \varphi/C_y \tag{13}$$

$$F_{y\beta r}l_r = \left(F_{y\beta fL} + F_{y\beta fR}\right)l_f$$
$$C_y\beta F_{zr}l_r = C_y\beta\left(F_{zfL} + F_{zfR}\right)l_f \tag{14}$$

$$F_{zr}/\left(F_{zfL} + F_{zfR}\right) = l_f/l_r \tag{15}$$

$x$: longitudinal direction
$y$: lateral direction
$v$: vehicle speed
$m$: vehicle mass
$g$: gravitational acceleration
$l$: wheel base
$Tr$: front tread
G.C.: gravity center
$l_f$: front distance from G.C.
$l_r$: rear distance from G.C.
$\varphi$: slant angle ($\approx$tilt angle)
$GCH$: gravity center height
$K_y$: cornering stiffness
$C_y$: normalized cornering stiffness

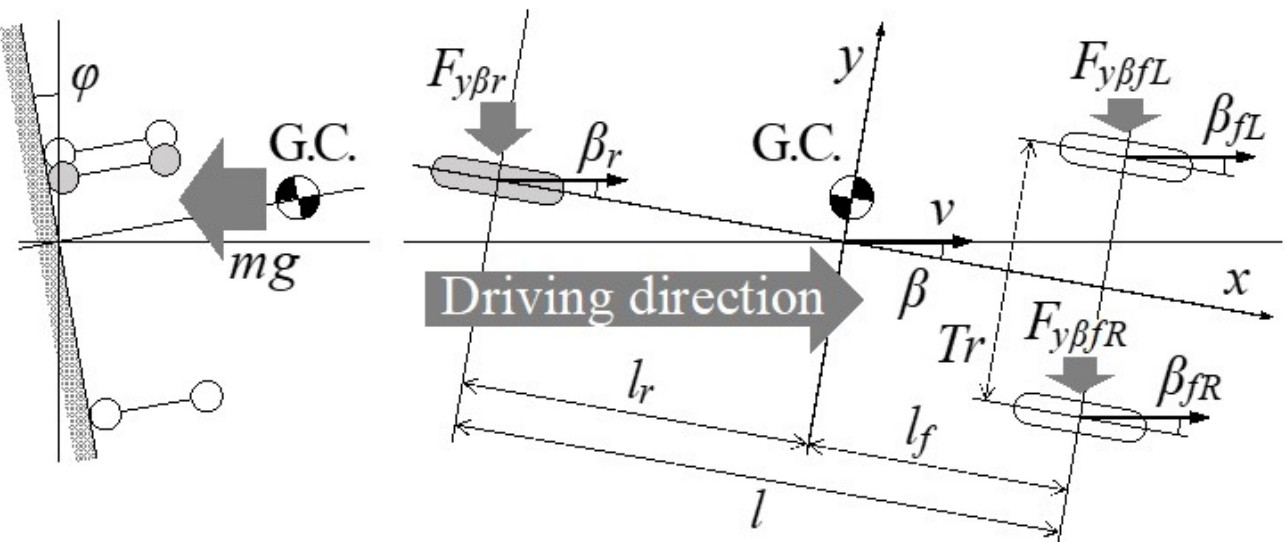

**Figure 14.** Straight running on a slanted road without inward tilting (car-like).

Next, consider a vehicle with an inward-tilting mechanism. According to the Reference [14], when the inward tilt angle of the motorcycle is just balanced with the roll moment during turning (Equation (16)), the lateral force due to the tire camber angle that accompanies the inward tilt is generally set to almost the value that it just balances the centrifugal force. This is not limited to the tires assumed in this report. As shown in Figure 15, when a PMV with an inward-tilting mechanism remains upright on a road surface with a slant angle, this relationship means that the lateral force generated by the camber angle of the tires prevents the vehicle from sliding down along the road surface. Therefore, the PMV stays running straight. At this time, there is no lateral transfer of vertical load due to the slant, no tire slip angle is required, unlike a vehicle without an inward-tilting mechanism, and no yaw angle is required for the vehicle body.

$$\varphi = tan^{-1}\left(v^2/\rho\right) \tag{16}$$

$$\begin{aligned}
\gamma_r &= \gamma_{fL} = \gamma_{fR} = \gamma = \varphi \; F_{y\gamma r} \approx Q_{yr}\gamma = D_y F_{zr}\gamma \\
F_{y\gamma fL} &\approx Q_{yfL}\gamma = D_y F_{zfL}\gamma \; F_{y\gamma fR} \approx Q_{yfR}\gamma = D_y F_{zfR}\gamma \\
mgsin\varphi &= F_y = F_{y\gamma r} + F_{y\gamma fL} + F_{y\gamma fR} = D_y\gamma\left(F_{zr} + F_{zfL} + F_{zfR}\right) = mgD_y\gamma
\end{aligned} \tag{17}$$

$$\gamma = \varphi \approx sin\varphi/D_y \; D_y \approx sin\varphi/\varphi \tag{18}$$

$$\begin{aligned}
F_{y\gamma r}l_r &= \left(F_{y\gamma fL} + F_{y\gamma fR}\right)l_f \\
D_y\gamma F_{zr}l_r &= D_y\gamma\left(F_{zfL} + F_{zfR}\right)l_f
\end{aligned} \tag{19}$$

$Q_y$: camber stiffness
$D_y$: normalized camber stiffness

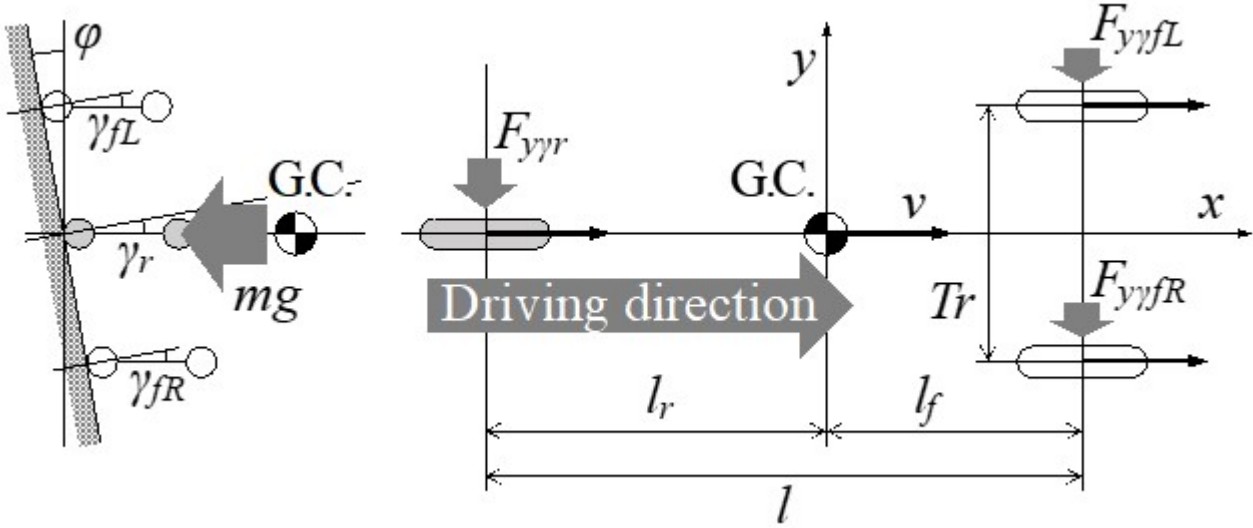

**Figure 15.** Straight running on slanted road with inward tilting (MC-like).

Figure 15 simply shows the balance when traveling straight on a slanted road, ignoring the lateral movement of the tire contact point due to the camber angle ($\gamma$) against the road surface. The lateral force balance is expressed by Equation (17), and the normalized camber stiffness ($D_y$) required to realize the state shown in Figure 3 is obtained by Equation (18). The balance of the yaw moment is expressed by equation (19), but this equation is nothing more than the indication of the vehicle's center of gravity (GC) position (Equation (15)). This relationship is equivalent to the balanced state of a motorcycle running straight on a slanted road.

Figure 16 shows the details of one wheel in Figure 15 as viewed from the rear, and considers the lateral movement of the tire contact point due to the camber angle against road surface. The amount of lateral movement is expressed as $CRsin\varphi$ using the tire crown

radius. In order to offset the turnover moment in the roll direction, the vehicle leans slightly to the mountain side. Therefore, the camber angle ($\gamma$) against the road surface is slightly larger than the slant angle ($\varphi$) by $\Delta\varphi$ as shown in Equation (20).

$$\gamma = \varphi + \Delta\varphi = \varphi + tan^{-1}(CRsin\varphi/GCH) \tag{20}$$

$$F_{y\gamma} = (\gamma + \Delta\gamma)D_yF_z{}^* = (\varphi + \Delta\varphi)D_ymgcos\varphi = mgsin\varphi \tag{21}$$

$$D_y = mgsin\varphi/((\varphi + \Delta\varphi)mgcos\varphi) = tan\varphi/(\varphi + \Delta\varphi) \tag{22}$$

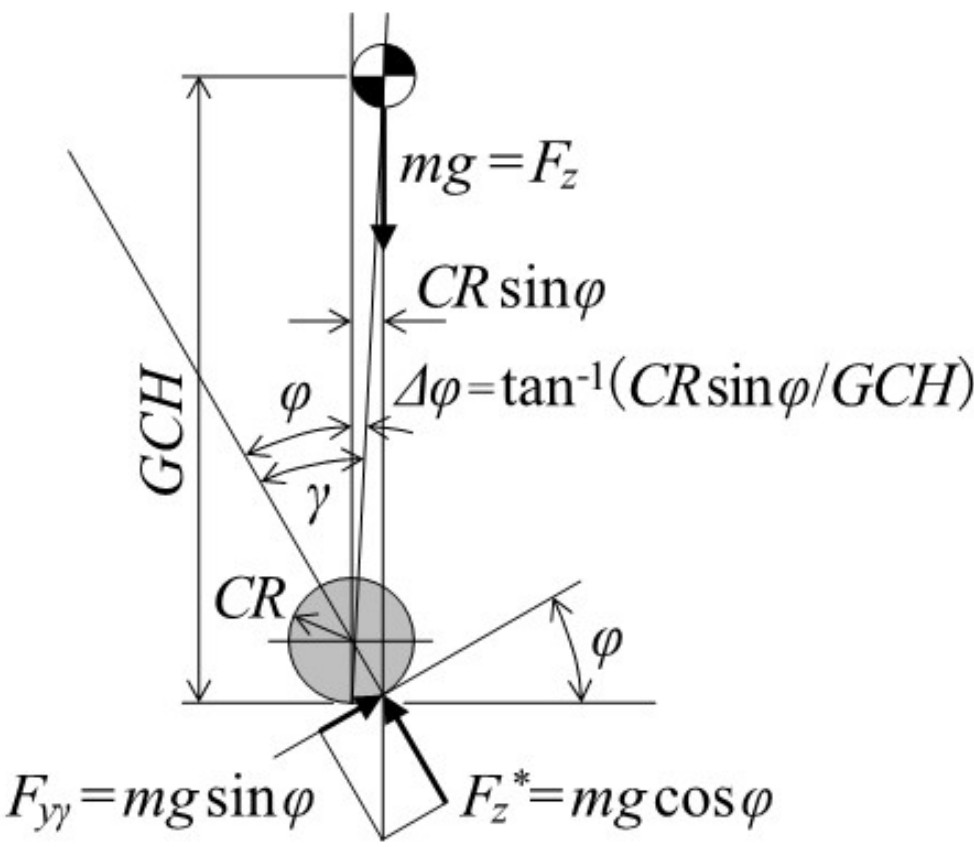

**Figure 16.** Lateral movement of the grounding point on a slanted road.

Equation (21) must hold in order to balance the lateral forces at zero slip angle ($\beta$). By modifying Equation (21), the normalized camber stiffness ($D_y$) required to maintain straight running on a slanted road is obtained by Equation (22). Using the same tire crown radius ($CR$ = 57mm) as in Reference (15), the necessary normalized camber stiffness ($D_y$) for the slant angle ($\varphi$) is almost constant as shown in Figure 17. When $\varphi$ = 10deg, $D_y = 1.522 \times 10^{-2}$/deg, and this value is almost equal to the value of $D_y$ required for turning only with camber stiffness ($Q_y$), which is obtained in Reference [14]. In other words, for a PMV that tilts inward when turning, by giving appropriate tire characteristics, it is possible to achieve both turning characteristics that do not produce a slip angle ($\beta$) during turns and straight running ability on slanted roads.

### 4.2.2. Straight Running Stability on Slanted Road

In order for the vehicle to maintain straight running without being affected by road surface disturbances, as discussed in References [17–19], it is required that lateral force is balanced without the steering operation, the yaw moment does not occur, and the steering moment also does not occur. Figure 18 shows the details of the caster trail changes in the side view due to the movement of the tire contact point, in addition to the details of

the rear view of the tire. Since $\Delta\varphi$ is small, the tire is assumed circular in the side view. The effective caster trail ($T_\xi^*$) is slightly smaller than the caster trail ($T_\xi$) as a suspension specification because the tire contact point is slightly higher than the original position due to the lateral movement of the tire contact point. Expressed as an equation, it is smaller by $CR(1-\cos\varphi)\sin\xi$ as shown in Equation (23), however, the difference is very small that it can be ignored compared to $T_\xi$ as shown in Figure 19.

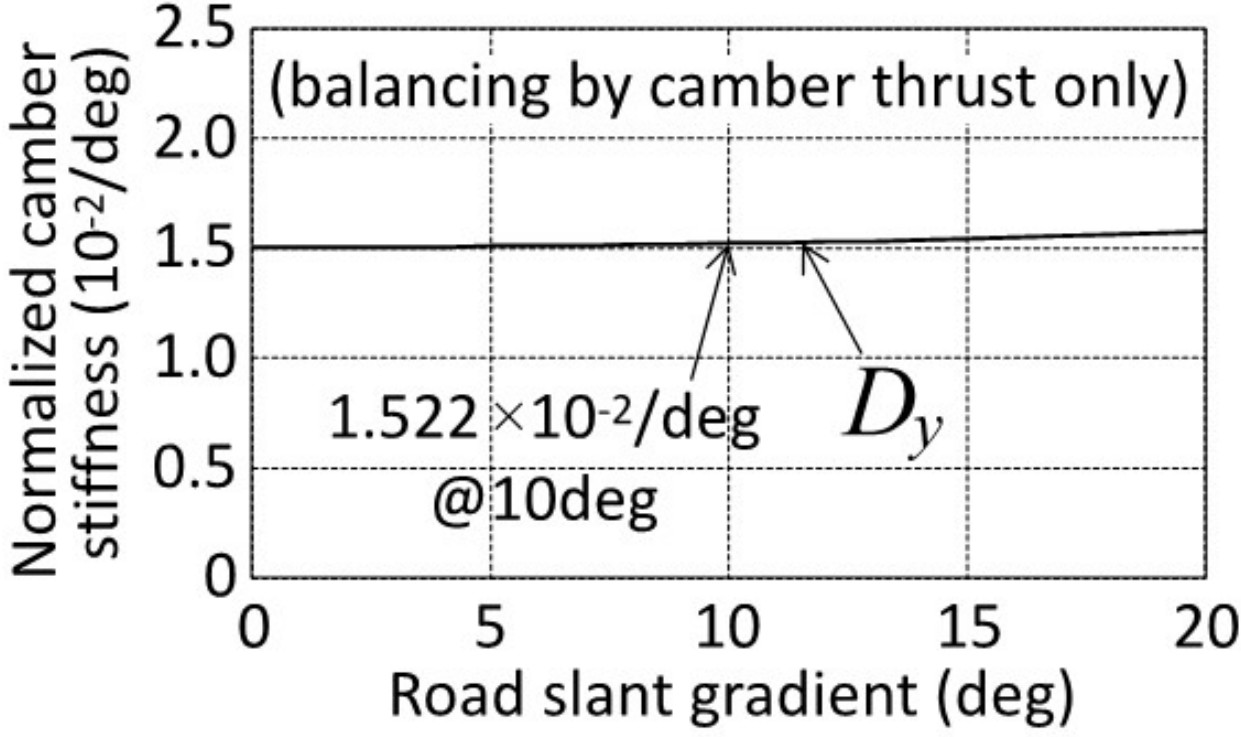

**Figure 17.** Normalized camber stiffness to cancel the road slant.

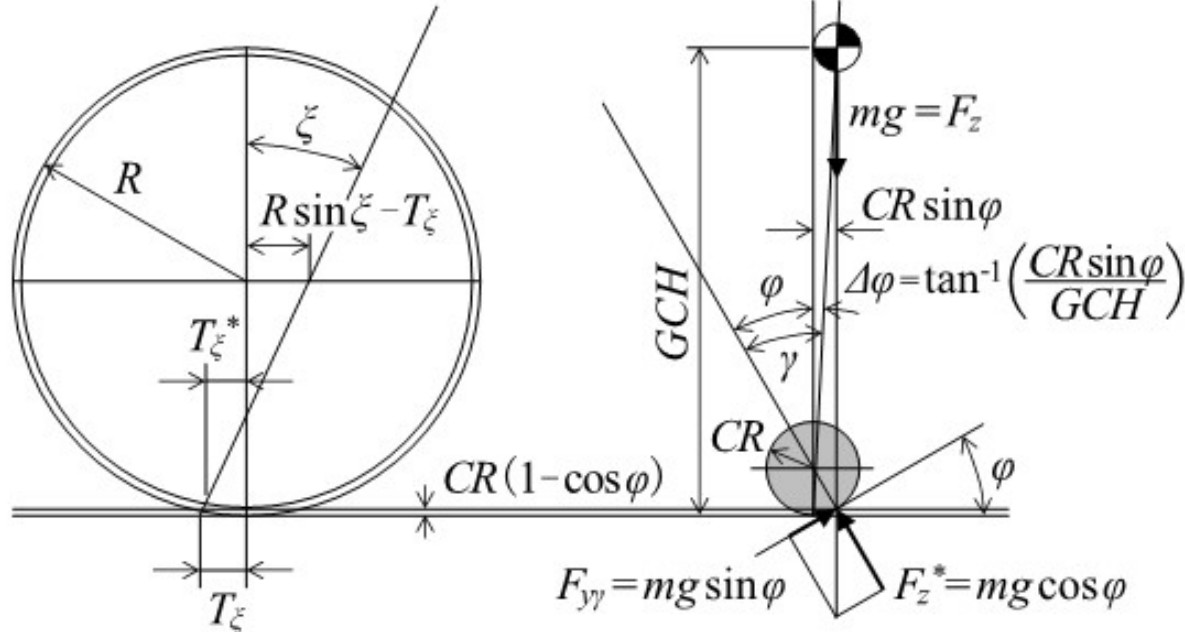

**Figure 18.** Steering geometry on a slanted road.

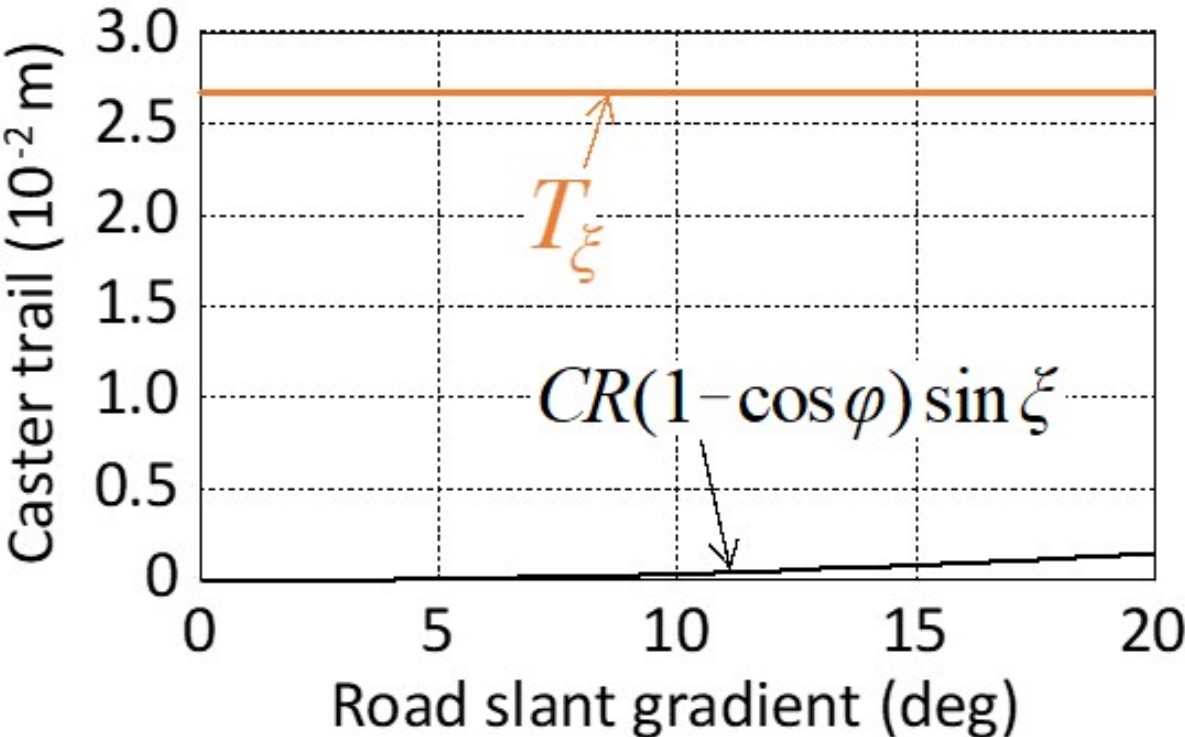

**Figure 19.** Reduced caster trail on a slanted road.

The moment around the steering axis consists of the sum of the moments by vertical load against road surface and the lateral force along with the road surface. The moment by the vertical load ($F_z{}^*$) against the road surface is equal to the moment around the steering axis when tilts are inward due to the reaction force of the vertical load ($F_z$) in Section 4.1.1. This is already zero, since the steering axis is followed the specification that satisfies the minimization requirement in Section 4.1.1. The steering moment ($M_z{}^*$) around the steering axis due to the lateral force is given by Equation (24). Since $CR(1 - \cos\varphi)\sin\xi$ is negligibly small compared to $T_\xi$, $M_z{}^*$ is effectively zero. In other words, as long as the concept of the steering axis setting shown in Section 4.1.2 is followed, when running on a transverse slanted road, the steering moment is not generated at the same time as the lateral force balance and the yaw moment balance.

$$T_\xi^* = (R - CR(1 - cos\varphi))sin\xi - \left(Rsin\xi - T_\xi\right) = T_\xi - CR(1 - cos\varphi)sin\xi \qquad (23)$$

$$M_z^* = F_{y\gamma}\left(e_\gamma + T_\xi^*\right)cos\xi cos\psi = mgsin\varphi cos\xi cos\psi\left(e_\gamma + T_\xi - CR(1 - cos\varphi)sin\xi\right) \qquad (24)$$

Unlike a vehicle without an inward-tilting mechanism, the vehicle body remains upright, therefore, only the unsprung part is involved in the dynamic response in the transition section, and the vehicle can move to the balanced state on the transverse slant part more quickly than a vehicle without an inward-tilting mechanism. However, the dynamic slant angle changes produce the significant vertical load change. As described in Section 4.1, if no moment is generated around the steering axis regardless of vertical load fluctuations, disturbance is minimized even in the slant angle transition section. Therefore, it can be said that the straight running ability of PMVs is extremely good.

4.2.3. Straight Running Stability on Rutted Road

In the actual market, the transverse slant of the road surface (Figure 20a) is a typical disturbance, but the rut is also a typical disturbance [17,18]. As shown in Figure 20b–d, in the case of ruts, unlike the case of slant, there is no particular difference in the height of

both wheels. As for the ruts, it is necessary to divide them into several cases, such as the difference in tread and the ruts formed by the twin tires of a large vehicle. When a PMV with a narrow tread runs on a rutted road formed by a passenger car or a large vehicle, various inconsistent situations can be assumed, such as the tread width of the PMV is close to the tread width of the rut and both wheels are positioned on the same side of the rut as shown in Figure 20b, such as the tread width of the PMV is narrower than the tread width of the ruts of a passenger car, and only one wheel is affected by the ruts as shown in Figure 20c, and such as the tread width of the PMV is fairly narrow and both wheels are positioned on symmetrical slopes on the even narrower ruts formed by the twin tires of large vehicles, as shown in Figure 20d.

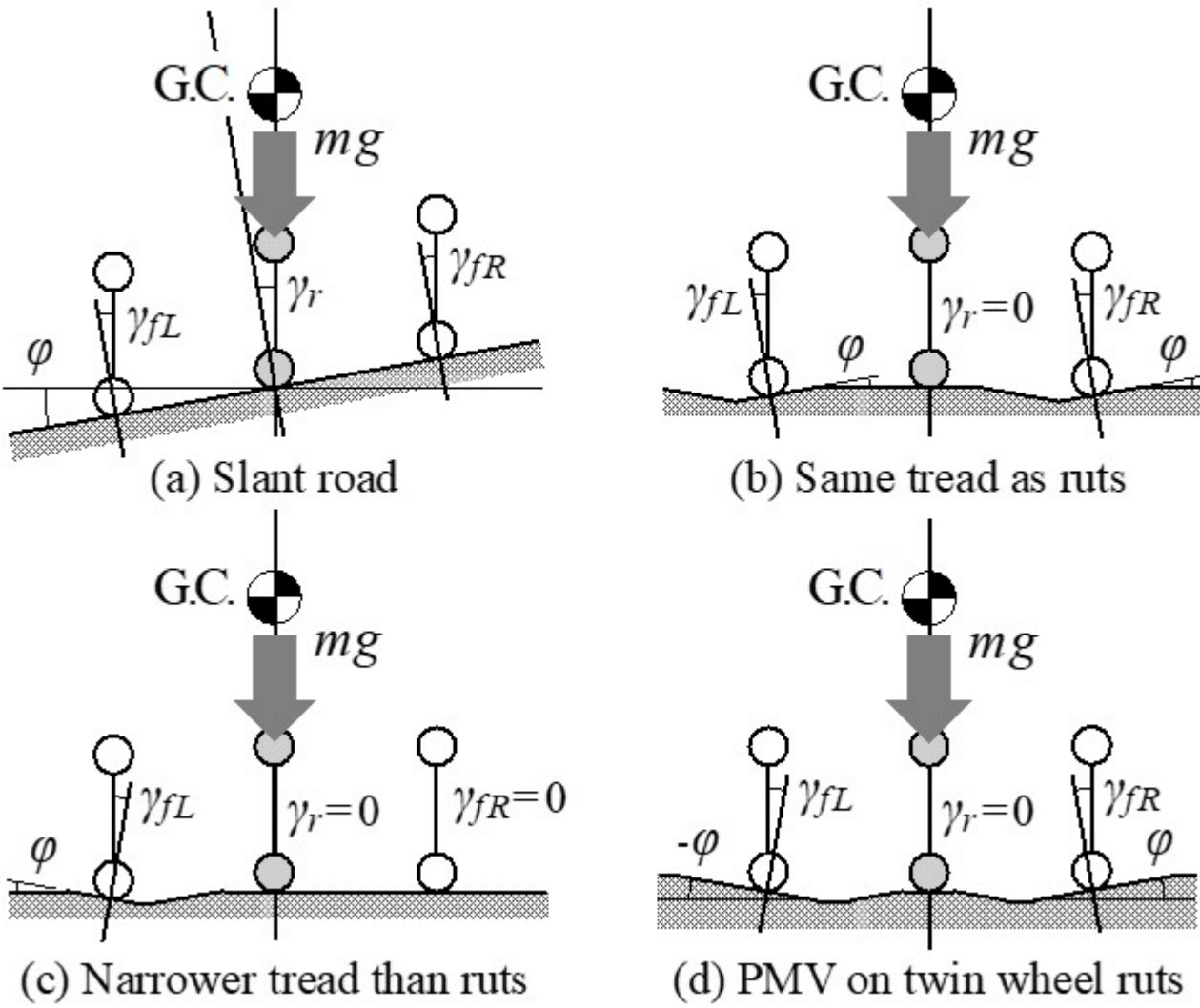

**Figure 20.** PMV running on a slanted road and typical rutted roads.

In order to maintain straight running without being disturbed by the transverse slope of the road surface in any of these inconsistent situations, the lateral force generated by the transverse slope of the road surface and the moment around the steering axis should not be canceled between the left and right wheels and lateral force and the moment must be avoided independently in each wheel. As mentioned in the discussion of straight running ability on slanted roads, if each of the both wheels independently minimize disturbances in terms of both the lateral force on slanted roads and the moment around the steering axis, straight running ability is maintained even under all ruts condition.

## 5. Discussion (Insight into Dynamic Phenomenon Analysis for the Future)

In this report, authors derived static balance equations for PMVs to be unaffected by road surface disturbances under the preconditions described in Section 3.1. The unsprung mass of the vehicle is not defined, nor is the delay in tire force/moment considered. Although these equations are statically the simulation model itself, they cannot analyze the dynamic behavior of PMVs. Dynamic behavior is expressed by a differential equation on the time axis, based on the acceleration of the vehicle body caused by static imbalance. As in this report, given static conditions that are unaffected by road surface disturbances in all situations, under the preconditions, the vehicle does not generate any acceleration, even dynamically.

This indicates that there is no point in constructing a dynamic model under the assumptions of this paper. Therefore, it is necessary to consider how to extend the preconditions of this report for future dynamic analysis.

- The dynamic behavior (1 to 2 Hz) of the entire vehicle is considered to be an accumulation of static balances, so in principle it is not affected by disturbances. Therefore, verification in this frequency domain should be entrusted to the confirmation phase with real vehicles.

- The possibility of vehicle response that cannot be explained by static balance due to transient characteristics of tire force and moment (affected by running speed, but about 5Hz at 36km/h, for example), which are not included in the assumptions of this report. Considering the response frequency of the vehicle, the transient vehicle response becomes a point of interest. Therefore, analysis with a dynamic model of the vehicle that incorporates a dynamic tire model is required. The authors are preparing this type of analysis as the next step of the research.

- Unsprung vibrations (approximately 10 Hz or higher), which are not considered in this report, may be transmitted to the entire vehicle in a vibratory manner. In order to reproduce even unsprung vibration, a dynamic model that considers the mass, inertia, and rigidity of each part of the vehicle, including the unsprung mass, is required. Furthermore, at practical speeds (e.g., 72 km/h), the transient characteristics of tires overlap with the frequency range, so analysis using a fairly advanced multi-degree-of-freedom model is required. Analysis of the steering axis arrangement from this point of view is difficult in the short term, and at the same time, it seems to deviate from the research scope of straight running performance of the vehicle on slanted roads and rutted roads.

In the future, the authors will proceed with their study on dynamic vehicle behavior based on the above considerations. Prior to the driving tests on prototype vehicles, we will use a multibody dynamics (MBD) model to directly verify the mechanical considerations in this report, and then proceed with analysis including tire transient characteristics. Going forward, we will continue our activities toward the realization of ultra-compact PMVs by making use of the knowledge obtained in this study.

## 6. Conclusions

In this paper, we studied the stability of the vehicle when external disturbances emanate from the road surface, such as left-right unequal changes in reaction force against the vertical load (impact from the road surface) due to road surface unevenness, and the transverse inclination of road surface due to slant and ruts. Then we derived the design requirements for the steering axis of PMVs to maintain straight running ability against such disturbances as follows.

- Based on the characteristics of general motorcycle tires, the centripetal force required for turning is obtained mainly by the camber angle of the tires to the ground (camber thrust). This means the lateral force of each tire is balanced on every angle of transverse slant as mentioned in Section 3.1.3. Therefore, steering torque is focused in order to avoid the disturbances on uneven road surface.

- Four unknown steering axis design parameters (caster angle ($\zeta$), caster trail ($T_\zeta$), kingpin angle ($\psi$), kingpin offset ($D_\psi$)) to derive in order to minimize the steering torque disturbances, are reduced into two unknowns (caster angle ($\zeta$), kingpin angle ($\psi$)) by previous studies of requirements for caster trail ($T_\zeta$) considering tire lateral force and kingpin offset ($D_\psi$), requirements considering the longitudinal braking force as mentioned in Section 3.2.2.
- Two unknown steering axis design parameters (caster angle ($\zeta$), kingpin angle ($\psi$)) are derived from two vehicle requirements on steady characteristics, as minimization of steering disturbance due to reaction force against the vertical load when standing upright and maintaining zero steering disturbance even if the tire contact point moves laterally when the vehicle is tilted inward, as mentioned in Section 4.1. This derivation is a new knowledge from a completely unique point of view.
- The lateral force balance and minimized steering torque free from the disturbance of each wheel on transverse slant of road surface as mentioned in Sections 4.2.1 and 4.2.2. Then the free from the disturbance of each wheel on transverse slant gives free from every kind of rut of road surface that is the combination of various transverse slant angles as mentioned in Section 4.2.3.

In this study, it was proved that the steering axis parameters can be derived uniquely by taking into consideration the requirement to zero the moment (disturbance) around the steering axis due to the reaction force against the vertical load at all internal tilt angles.

**Author Contributions:** Conceptualization, T.H.; methodology, T.H.; software, T.H. and T.K.; validation, T.H.; formal analysis, T.H.; investigation, T.H. and T.K.; resources, T.H. and T.K.; data curation, T.H.; writing—original draft preparation, T.H.; writing—review and editing, T.H.; visualization, T.H.; supervision, T.H.; project administration, T.H.; funding acquisition, T.H. and T.K. All authors have read and agreed to the published version of the manuscript.

**Funding:** This research received no external funding.

**Data Availability Statement:** Not applicable.

**Conflicts of Interest:** The authors declare no conflict of interest.

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
