# Peer review of "Design Requirements for Personal Mobility Vehicle (PMV) with Inward Tilt Mechanism to Minimize Steering Disturbances Caused by Uneven Road Surface"

_inventions, doi:10.3390/inventions8010037_

Round 1
Reviewer 1 Report
First of all, I want to congratulate the authors for their efforts in this manuscript. The paper proposed a new mechanism to reduce the Steering Disturbances Caused by Uneven Road Surfaces. The topic is aligned with the journal’s scope and is interesting for the readers. There are some aspects to be improved before accepting the paper. One of the major limitations of the work is the lack of an extensive review of the existing similar experiments and the lack of context. Only 20 references have been cited so far. The other limitation is the lack of results that demonstrate the minimization of Steering Disturbances Caused by Uneven Road Surfaces. Following, I include the list of issues to be corrected:
English must be proofread in some sections, such as the conclusions, while others as the introduction, are correct.
Avoid using the same word in the keywords and in the title. Consider changing the underwater personal mobility vehicle by another keyword.
In the introduction, solutions to monitor the axis acceleration should be cited. Check this new reference on this topic (https://doi.org/10.3390/electronics11233965, check for other similar examples).
The aim of the paper, as well as the novelty of the proposal, must be clearly mentioned in the introduction. I suggest using bullet points.
The structure of the paper must be included after the aim of the paper at the end of the introduction.
A related work section should be included after the introduction; in this section, authors must summarize the state of art.
Results must be added to the paper. The authors must simulate or develop a test to demonstrate the benefits of the proposed mechanism. This is mandatory before accepting the paper.
Reviewer 2 Report
1. The innovation of this paper has been clearly stated in the last paragraph of the Introduction.
2. No quantified contributions are presented in Abstract and Conclusions.
3. The meaning of all figures and formulas shown in this paper should be explained detail, for example, the side slip angle and camber angle in Figure 3, the B1 and B2 in Figure 5.b, symbols in Equation (3) and Figure 7, and the solid and dashed lines in Figure 9.
4. Are there any hypotheses about the formulas |P1-P2|=1 and 0≦Lx, Ly, Lz≦1?
5. Page 5: …the References [12] [13], as shown in…->…the References [12, 13], as shown in…
6. Page 16: (c) PMV on twin wheel ruts -> (d) PMV on twin wheel ruts.
7. Are there any hypotheses about the formulas |P1-P2|=1 and 0≦Lx, Ly, Lz≦1?
8. Some cases are discussed in this article, for example, sudden changes in reaction force against vertical load and the transvers inclination of road surface due to slant and ruts. However, its impact on PMV must be emphasized and the plausibility must also be demonstrated.
9. The conclusions should be revised, and the major contributions should be shown.
Reviewer 3 Report
This research paper presents an interesting concept to minimize Steering Disturbances Caused by uneven road surfaces. Although a good Vehicle and Tire specifications deliberation is provided the introduction section needs to be consolidated to focus on the actual objectives of the research. Although the methodology is comprehensive and systematic, however, more discussion is needed to better explain the reduction guidelines of the proposed steering Disturbances testing mechanism. Finally, the conclusion section is too brief and needs to focus on the actual results and provide additional deliberations on the testing shortcomings and thus further research.
Round 2
Reviewer 1 Report
The authors have addressed the comments.